# Perineal Wound Closure Following Abdominoperineal Resection and Pelvic Exenteration for Cancer: A Systematic Review and Meta-Analysis

**DOI:** 10.3390/cancers13040721

**Published:** 2021-02-10

**Authors:** Etienne Buscail, Cindy Canivet, Jason Shourick, Elodie Chantalat, Nicolas Carrere, Jean-Pierre Duffas, Antoine Philis, Emilie Berard, Louis Buscail, Laurent Ghouti, Benoit Chaput

**Affiliations:** 1Department of Digestive Surgery, Toulouse University Hospital, 31100 Toulouse, France; etienne.buscail@inserm.fr (E.B.); canivet.c@chu-toulouse.fr (C.C.); carrere.n@chu-toulouse.fr (N.C.); duffas.jp@chu-toulouse.fr (J.-P.D.); philis.a@chu-toulouse.fr (A.P.); ghouti.l@chu-toulouse.fr (L.G.); 2INSERM, U1220, Digestive Health Research Institute (IRSD), University of Toulouse, 31100 Toulouse, France; 3Department of Gastroenterology and Pancreatology, Toulouse University Hospital, 31100 Toulouse, France; 4Department of Epidemiology and Public Health, UMR 1027 INSERM, Toulouse University Hospital, University of Toulouse, 31100 Toulouse, France; usmr@chu-toulouse.fr (J.S.); emilie.berard@univ-tlse3.fr (E.B.); 5Department of Surgery, Oncopole, INSERM-UPS UMR U1048, Institute of Metabolic and Cardiovascular Diseases, University of Toulouse, 31100 Toulouse, France; chantalat.e@chu-toulouse.fr; 6Department of Plastic and Reconstructive Surgery, Toulouse University Hospital, 31100 Toulouse, France; chaput.b@chu-toulouse.fr

**Keywords:** rectal cancer, abdominoperineal resection, flap, mesh, perineal wound healing, perineal morbidity, surgical oncology

## Abstract

**Simple Summary:**

Abdominoperineal resection (APR) and pelvic exenteration (PE) for the treatment of cancer (mainly anal and rectal cancers) require extensive pelvic resection with a high rate of postoperative complications. The objective of this work was to systematically review and meta-analyze the effects of vertical rectus abdominis myocutaneous flap (VRAMf) and mesh closure on perineal morbidity following APR and PE. The studies were distributed as follows: Group A comparing primary closure (PC) and VRAMf, Group B comparing PC and mesh closure, Group C comparing PC and VRAMf in PE. The meta-analysis of Groups A and B showed PC to be associated with an increase in the rate of total and major perineal wound complications. PC was associated with a decrease in total and major perineal complications in Group C.

**Abstract:**

Background. Abdominoperineal resection (APR) and pelvic exenteration (PE) for the treatment of cancer require extensive pelvic resection with a high rate of postoperative complications. The objective of this work was to systematically review and meta-analyze the effects of vertical rectus abdominis myocutaneous flap (VRAMf) and mesh closure on perineal morbidity following APR and PE (mainly for anal and rectal cancers). Methods. We searched PubMed, Cochrane, and EMBASE for eligible studies as of the year 2000. After data extraction, a meta-analysis was performed to compare perineal wound morbidity. The studies were distributed as follows: Group A comparing primary closure (PC) and VRAMf, Group B comparing PC and mesh closure, and Group C comparing PC and VRAMf in PE. Results. Our systematic review yielded 18 eligible studies involving 2180 patients (1206 primary closures, 647 flap closures, 327 mesh closures). The meta-analysis of Groups A and B showed PC to be associated with an increase in the rate of total (Group A: OR 0.55, 95% CI 0.43–0.71; *p* < 0.01/Group B: OR 0.54, CI 0.17–1.68; *p* = 0.18) and major perineal wound complications (Group A: OR 0.49, 95% CI 0.35–0.68; *p* < 0.001/Group B: OR 0.38, 95% CI 0.12–1.17; *p* < 0.01). PC was associated with a decrease in total (OR 2.46, 95% CI 1.39–4.35; *p* < 0.01) and major (OR 1.67, 95% CI 0.90–3.08; *p* = 0.1) perineal complications in Group C. Conclusions. Our results confirm the contribution of the VRAMf in reducing major complications in APR. Similarly, biological prostheses offer an interesting alternative in pelvic reconstruction. For PE, an adapted reconstruction must be proposed with specialized expertise.

## 1. Introduction

Perineal wound problems after abdominoperineal resection (APR) in the context of anal and rectal cancers are frequent [1]. These types of resection problems occur because of wound complications caused by large perineal defects. Indeed, perineal wound complications, perineal abscesses, wound dehiscence, chronic fistulas, and sinuses lengthen hospital stays. Furthermore, the standardization of the surgery since the late 2000s and the extra-levator abdomino-perineal resection [2] (ELAPE) technique have led to a larger defect and increased perineal complications. In addition, advances in terms of pelvic oncology make it possible to manage increasingly advanced diseases or cancer recurrence [3], leading to the performance of pelvic exenterations [4].

Several strategies are used to decrease the complication rate. Closure by direct approximation of the pelvic muscles leads to a rate of major complications up to 57%, depending on the series [5]. Musculocutaneous flaps help to reduce this rate (16–46%) [6,7], but they generate their own morbidity, require experience, and require a specialized reconstructive surgery team. Finally, since the early 2010s, the use of biologic meshes seems to have improved the healing process [8,9]. However, results are still variable, and a recent randomized controlled study comparing direct closure and mesh closure showed no significant difference on morbidity at one year [10].

The effectiveness of flap and mesh closure strategies remains uncertain due to the small size of the populations studied, the heterogeneity of studies, and the varying availability of the data and experience of reconstructive surgeons in different institutions [11,12]. We conducted a systematic review and meta-analysis to compare myocutaneous flaps versus primary closure, as well as mesh closure versus primary closure for pelvic reconstruction after abdominal perineal resection and pelvic exenteration, with regard to perineal complications and post-operative length of stay.

## 2. Materials and Methods

The study protocol was prospectively registered at PROSPERO (registration number: CRD42020185719), following the Preferred Reporting Items for Systematic Reviews and Meta-analysis (PRISMA), and Meta-Analyses of Observational Studies in Epidemiology (MOOSE) checklist [13,14].

### 2.1. Search Strategy and Data Collection

All studies reporting on perineal wound healing after abdominoperineal resection for rectal and anal cancer were considered eligible for analysis. The literature was systematically reviewed by searching in the PubMed library, EMBASE, and Cochrane library from 2000 to March 2020. The used medical subject headings (MESH) used are summarized in Appendix A.

### 2.2. Inclusion and Exclusion Criteria

Original studies aiming at comparing the surgical outcomes of flap and primary closure or mesh and primary closure after abdominoperineal resection and pelvic exenteration for malignant indication were included. Exenteration is defined as the excision of the tumor mass (including rectum or neorectum) and at least one adjacent organ. Randomized controlled trials and observational studies including patients undergoing abdominoperineal resection for cancer and reporting primary, flap, and mesh closure were potentially eligible.

Articles meeting the following criteria were excluded: case series, case reports, commentaries, letters to the editor, literature reviews or meta-analyses, non-English language, cohort less than 10 patients, inflammatory bowel disease studies, or surgical intervention or outcomes of interest not reported. Studies evaluating only a single intervention group were excluded. Studies that did not report details of complications and those eliciting no response from the corresponding authors contacted by mail were also excluded.

### 2.3. Outcome Parameters

Our study analysis compared outcome settings defining perineal wound complications of flap, primary, and mesh closure procedures. This included total perineal wound complications. Major and minor perineal wound complications were classified according to the Southampton Wound Assessment Scale [10,15] (Appendix A). When the Southampton score was not established in a study, complications were classified according to the details given and the corresponding authors’ answers. A score of 0 or 1 reports normal healing, a score of 2 or 3 reports minor complications, and a score of 4 or 5 reports major complications. Studies were eligible if one or more of these outcomes were reported or provided by the corresponding author. The length of postoperative stay was also collected. To reach optimal homogeneity and comparability, we divided the studies into three groups as follows: Group A studies comparing primary and flap closure in APR and ELAPE, Group B studies comparing primary and mesh closure in in APR and ELAPE, and Group C studies comparing primary and flap closure in pelvic exenterations only.

### 2.4. Data Collection and Extraction

A data abstraction form was designed a priori with Excel (Microsoft Corp, Redmond Washington, DC, USA) to standardize data collection. Two independent reviewers (E. Buscail. and C.Ca.) scanned all abstracts identified by search cross-referencing. The full text was then identified for each study that potentially met the inclusion criteria. Two reviewers (E. Buscail and C. Canivet) independently reviewed the full-text eligibility. If no consensus could be reached by the two reviewers and after discussion, a third specialist author was consulted and made the decision (B. Chaput). Articles that did not meet the inclusion criteria were excluded (Figure 1). Data extraction included general study information (age, sex ratio, BMI), neoadjuvant treatment (long/short-course radiotherapy and chemotherapy), surgical data (surgical resection technique, perineal reconstruction type), postoperative perineal wound healing details (total, major, and minor complication, and flap failure), and the postoperative length of stay and the duration of follow-up. In case of missing data or lack of details regarding the study, the corresponding authors were contacted. 

### 2.5. Quality Assessment

A quality assessment of the studies was performed by two independent reviewers (E. Buscail and C. Canivet) using the Newcastle-Ottawa Scale [16,17]. The study quality was evaluated in 3 domains: patient selection, comparability, and outcome. A star rating of 0 to 9 was allocated to each study according to these parameters. Studies with scores of 8 or higher were classified as high quality, scores of 4 to 8 as moderate quality and scores of less than 4 as low quality. Any disagreement was resolved by discussion or by consulting a third investigator (B. Chaput).

### 2.6. Statistical Analysis

Each group was described using a weighted incidence rate for total, major, and minor complications, and a weighted pooled mean for length of stay. Overall odds ratio (OR) was used to compare complication rates and standardized mean difference (SMD) for length of stay. The generic inverse variance method was used to weight the effects. We assessed statistical heterogeneity between trials using the I^2^ statistic. We used a fixed effect model if I^2^ was below 0.5 and a random effects model if it was above 0.5. We evaluated the potential publication bias using funnel plot analysis and Egger’s test for each comparison of the total complication rate. All analyses were conducted using R statistical software version 4 and the metafor package. Significance was set for values <0.05.

## 3. Results

### 3.1. Literature Search and Results

The initial search resulted in 2588 citations, and one citation was added with a manual cross-reference. Then, 2088 titles remained after removing duplicates. After screening, 71 publications were retrieved for full-text review. When data was not available, the corresponding authors were contacted: four of them answered [18,19,20,21], and two of them could provide data details [18,21]. From the 71 articles, 53 articles were excluded, and the reasons are detailed in Figure 1. Finally, eighteen articles were included in quantitative synthesis (Figure 1) [6,7,9,10,18,21,22,23,24,25,26,27,28,29,30,31,32,33].

### 3.2. Study and Patient Characteristics

Study descriptions are provided in Table 1. Study endpoint details for each study are provided in the Appendix A. The quality of studies was moderate to good (range 3–9; Table 1 and Appendix A). Fourteen compared flap to primary closure, including three studies on pelvic exenteration, and four articles compared mesh to primary closure. The 18 included studies covered 2180 patients (1206 primary closures, 647 flap closures, and 327 mesh closures). Pooled baseline characteristics are displayed in Table 2.

### 3.3. Study Endpoints

The outcomes of each study are detailed in the Appendix A. The main findings of the studies are summarized in Table 2. Visual inspection of the funnel plots for the main and secondary outcomes of interest, as well as the non-statistically significant Egger’s test for each comparison of the main outcome (*p* = 0.68 and *p* = 0.96), suggested the absence of publication bias in our study (Appendix A).

### 3.4. Total Perineal Wound Complication

Eighteen included studies reported total perineal wound complications [6,7,9,10,18,21,22,23,24,25,26,27,28,29,30,31,32,33]. Eleven studies reported total perineal wound complication in Group A [6,7,21,22,23,24,25,26,27,28,29,30] four studies in Group B [9,10,18,31], and three in Group C [25,32,33]. The overall pooled weighted rate for the primary closure was 39.9% (95% CI 35.7%–44.6%) in Group A, 30.6% (95% CI 23.3%–40%) in Group B, and 51.3 % (95% CI 33.3%–79.6%) in Group C. The overall pooled weighted rate for flap closure was 30.3% (95% CI 21.8 %-42.2%) in Group A and 67.2% (95% CI 52.2%–86.5%) in Group C. The overall pooled weighted rate for mesh closure (Group B) was 18.7% (95% CI 10.6%–32.9%). Meta-analysis shows that primary closure is associated with a significant increase in total perineal wound complications compared to flap closure in Group A (OR 0.55, 95% CI 0.43–0.71; *p* < 0.01; I^2^ = 34%; *p* < 0.01). In Group B, the primary closure is associated with a higher rate of total perineal wound complication, but this result was not significant (OR 0.54, CI 0.17–1.68; *p* = 0.18; I^2^ = 54%; *p* = 0.18). In Group C, flap closure is significantly associated with an increase in total perineal complications (OR 2.46, 95% CI 1.39–4.35; *p* < 0.01; I^2^ = 46%). (Figure 2).

### 3.5. Major Wound Complications

Sixteen studies reported major perineal wound complications [6,7,9,10,18,22,23,25,26,27,28,29,30,31,32,33]. Ten studies reported major perineal wound complication rates in Group A [6,7,22,23,25,26,27,27,29,30], four studies in Group B [9,10,18,31], and three in Group C [25,32,33]. The overall pooled weighted rate for the primary closure was 23.6% (95% CI 17%–32.8%) in Group A, 20.7% (95% CI 14.8%–28.8%) in Group B, and 27.6% (95% CI 11.2%–68%) in Group C. The overall pooled weighted rate for flap closure was 14.3% (95% CI 11.2 %–18.3%) in Group A and 36.7% (95% CI 20.7%–65.3%) in Group C. The overall pooled weighted rate for mesh closure (Group B) was 10.5% (95% CI 7.3%–15.2%).

Meta-analysis shows that primary closure is associated with a significant increase in major perineal wound complications compared to flap closure in Group A (OR 0.49, 95% CI 0.35–0.68; *p* < 0.001). In Group B, major complications ranged significantly higher in primary closure (OR 0.38, 95% CI 0.12–1.17; *p* < 0.01). In Group C, flap closure is associated with an increase in major perineal complications (OR 1.67, 95% CI 0.90–3.08; *p* = 0.1). A low heterogeneity was identified for reported events in each group (I^2^ Group A = 22%; I^2^ Group B = 24%; I^2^ Group C = 0%) (Figure 3).

### 3.6. Minor Wound Complications

Fifteen studies reported minor perineal wound complications [6,7,9,10,18,22,23,25,26,28,29,30,31,32,33]. Nine studies reported minor perineal wound complications in Group A [6,7,22,23,25,26,28,29,30], four in Group B [9,10,18,31], and three in Group C [25,32,33]. The overall pooled weighted rate for the primary closure was 20.9% (95% CI 17.7%–24.6%) in Group A, 13% (95% CI 7.8%–21.5%) in Group B, and 15.6% (95% CI 11.1%–21.9%) in Group C. The overall pooled weighted rate for flap closure was 18.4% (95% CI 10.5 %–32.4%) in t Group A and 32.2% (95% CI 13.3%–77.8%) in Group C. The overall pooled weighted rate for mesh closure (Group B) was 13.9% (95% CI 6.5%–26.9%). Meta-analysis did not show any difference in studies comparing flap and primary closure: Group A (OR 0.83; 95% CI 0.39–1.79; *p* = 0.60) and Group C (OR; 95% CI 0.04–135.57; *p* = 0.47), with high heterogeneity (I^2^ = 70% and I^2^ = 86%, respectively). There was also no difference in the study group comparing the meshes to the primary closure (Group B (OR 1.09; 95% CI 0.54–2.21; *p* = 0.80)) with a more limited heterogeneity (I^2^ = 41%) (Figure 4).

### 3.7. Length of Stay

Eight articles providing data for the postoperative length of stay in Group A were included in our meta-analysis [6,7,21,22,26,27,28,29]. The pooled weighted mean length of stay was 15.1 days (95% CI 13.2, 16.9) for primary closure, and 17.4 (95% CI 13.5, 21.2) days for flap closure, which was not significantly different (SMD −0.07 CI 95% −0.25–0.11) (Figure 5).

## 4. Discussion

Our literature review and meta-analysis showed the necessity to fill the vacuum left by APR to reduce major perineal wound complications. This observation is more difficult to assert for more extensive resections, which are most often for more specialized and less standardized indications such as recurrence of pelvic cancer [4] or rarer neoplasia such as sarcomas. For this reason, we have endeavored to separate the studies of pelvic exenterations, on the one hand, and to standardize the classification of complication outcomes with the Southampton Wound Complication Scale complications, on the other hand.

APR morbidity remains high. The series comparing flap closure and primary closure presented a rate for major perineal wound complications ranging from 13% to 55% [22,27] for primary closure and up to 26% for flap closure [6] (Appendix A). For studies comparing morbidity between mesh closure and primary closure, the major morbidity reported reaches 31% for primary closure [31]. The major complication rate is as high as 64% in pelvic exenterations [32]. This morbidity may be explained by the fact that the oncological results of a complete surgery are accompanied in the vast majority of cases by neoadjuvant radiotherapy, which is a risk factor for postoperative complications [5,34]. Another factor now being considered is the ELAPE, which is currently the standard for lower rectal cancer and leaves more dead space in the pelvis [35]. This space encourages the formation of collections, perineal abscesses, and chronic failure of the perineal wound [36,37].

Faced with the need to fill the space in the perineum to prevent the formation of collections, the option of musculocutaneous flaps has been increasingly used [38]. The most widely used and described here is the VRAM flap. A noteworthy result is that our meta-analysis shows a significant reduction in total (OR 0.55, 95% CI 0.43–0.71; I^2^ = 34%) and major complications (OR 0.49, 95% CI 0.35–0.68; *p* < 0.001) when using the VRAM flap. This result has already been shown in previous work, but the studies included gracilis flaps and pelvic exenterations [1]. The homogeneity of surgical strategies (e.g., VRAM flap and exclusion of pelvic exenteration) of the studies included here strengthens these findings. The flap donor site complications are one important aspect but are not discussed here, as we have focused our analyses on perineal wound healing.

The issue remains more complex and less clear-cut in the case of pelvic exenterations. Indeed, this specialized center surgery often requires a multidisciplinary team including plastic and pelvic surgeons [39]. This surgery is intended for patients with a long oncological history who have received numerous courses of chemotherapy and often even re-irradiation for pelvic recurrences [4]. Our meta-analysis shows a significant increase in total perineal complications in case of closure with the VRAM flap (OR 2.46, 95% CI 1.39–4.35; *p* < 0.01; I^2^ = 46%) This result is found for major complications, but it was not significant (OR 1.67, 95% CI 0.90–3.08; *p* = 0.1). Davidge et al. report a 4.8 higher perineal complication rate in cases of sacral resection manifested most often by dehiscence at the junction of the sacral incision and the posterior part of the flap [25]. Jacombs et al., who reported a retrospective study on 203 patients with pelvic exenteration, found a higher rate of major complications in patients with flap (25% vs. 14%) [33]. This result is weighted by the presence of risk factors identified by the authors: history of radiotherapy and APR, as well as total pelvic exenteration and sacral resection. Thus, the authors suggest that patients with fewer factors may have primary closure without increasing the rate of perineal wound complications.

More recently, closure with a biological prosthesis has emerged as an attractive alternative. It has the advantage that, unlike flaps, it does not require additional operating time and does not cause additional abdominal parietal defect. Interestingly, fewer major complications were found with the insertion of a prosthesis compared to primary closure (OR 0.38, 95% CI 0.12–1.17; *p* < 0.01). Of note, the meta-analysis includes two controlled prospective studies comparing primary closure with mesh closure. First, Munster et al. (BIOPEX study) [10] did not find any significant difference in total, minor and major complications. On the other hand, Han et al. found a significantly higher rate of total complications in the primary closure group than in the mesh closure group (12% vs. 36% *p* < 0.05) [31].

Besides the functional aspect, the medico-economic aspect could make the difference between the different closure strategies, but few studies report data on this subject. Woodfield et al. compared the direct costs and the costs of complications between a primary closure group (n = 37) and a VRAM flap closure group (n = 31) [29]. They reported higher costs in the VRAM group, particularly in relation to operating time and hospitalization; on the other hand, there were higher costs for the management of complications in the primary closure group (8394 New Zealand dollars vs. 25,911 New Zealand dollars, *p* = 0.012). Billig et al. compared a group of 2363 patients with primary closure and 194 patients with flap closure and found no significant difference between the two groups in terms of total and direct costs [40]. Regarding cost-effectiveness, it was higher for the primary closure group ($259/healthy day) than for the flap group ($186/healthy day) [40]; nevertheless, this result was not significant (*p* = 0.17). Currently, three ongoing multicenter trials could provide some answers as to which strategy to choose. The functional aspect was the question raised by the NEAPE trial (NCT 01347697) [41] comparing the gluteus flap to mesh closure. The issue of the gluteus flap is also studied by the BIOPEX-2 trial (NCT04004650) [42] with a primary closure control group. Finally, we are conducting an ongoing, randomized, multicenter, controlled trial on medico-economic issues and quality of life comparing primary closure with mesh closure by the GRECCAR 9 study (NCT 02841293).

Several limitations were encountered in this meta-analysis. First, this review included mostly observational studies, with only two randomized controlled trials and one controlled trial. Even with the strict inclusion criteria, limitations in the level of studies were met. Control groups (primary closure) were often poorly matched to study groups, especially for studies involving flaps, limiting the potency of the comparative analysis. Several studies used historical controls from an earlier period or constructed controls creating selection bias. Indications for perineal resection in groups A and C included patients with rarer cancers of the anus, rectum, and other neoplasias with less homogeneous neo-adjuvant treatments. However, our review was not significantly limited by the heterogeneity, as reflected by symmetric Funnel plots (Appendix A) and acceptable I^2^ value, except for minor perineal wound complication. In flap studies, patients with co-morbidities, especially cardiovascular disease (e.g., diabetes, arterial hypertension, etc.), are often excluded due to the risk of flap necrosis and are, therefore, in the primary closure control groups. However, this finding must be weighed against the fact that flap indications are often associated with larger resections.

## 5. Conclusions

Our meta-analysis is the first to include a comparative analysis of the two most commonly used strategies for perineal closure. Our results confirm the contribution of the VRAM flap in reducing major complications in non-extensive perineal resection. Similarly, biological prostheses offer an interesting alternative in pelvic reconstruction, especially in the case of laparoscopic reconstruction, allowing the abdominal wall to be spared. For pelvic exenterations, an adapted reconstruction must be proposed and requires specialized multidisciplinary collegial surgical expertise.

## Figures and Tables

**Figure 1 cancers-13-00721-f001:**
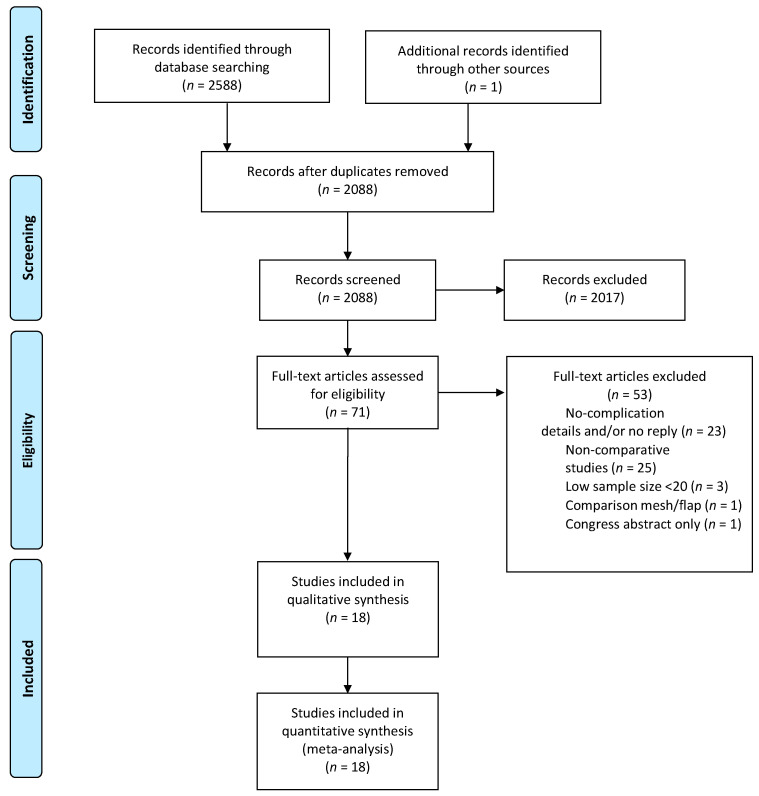
Flow diagram showing the search strategy for articles included in the systematic review and meta-analysis reported following the Preferred Reporting Systems for Systematic Reviews and Meta-Analysis statement.

**Figure 2 cancers-13-00721-f002:**
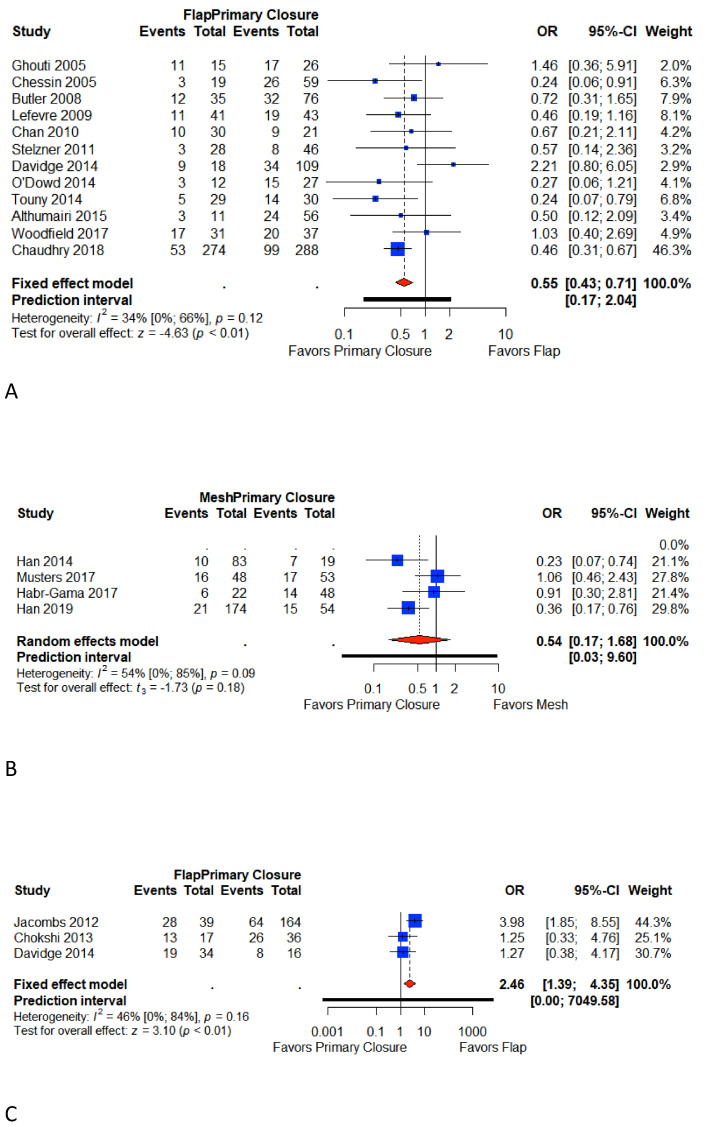
Forest plot for total perineal wound complications between patients with primary closure and with flap closure (**A**); forest plot for total perineal wound complications between patients with primary closure and with mesh closure (**B**); forest plot for total perineal wound complications between patients with primary closure and with flap closure in pelvic exenteration (**C**). Odds ratio: overall ratio of total perineal wound complications when comparing primary closure with flap closure. All weights are from random effects analysis. M–H, Mantel–Haenszel.

**Figure 3 cancers-13-00721-f003:**
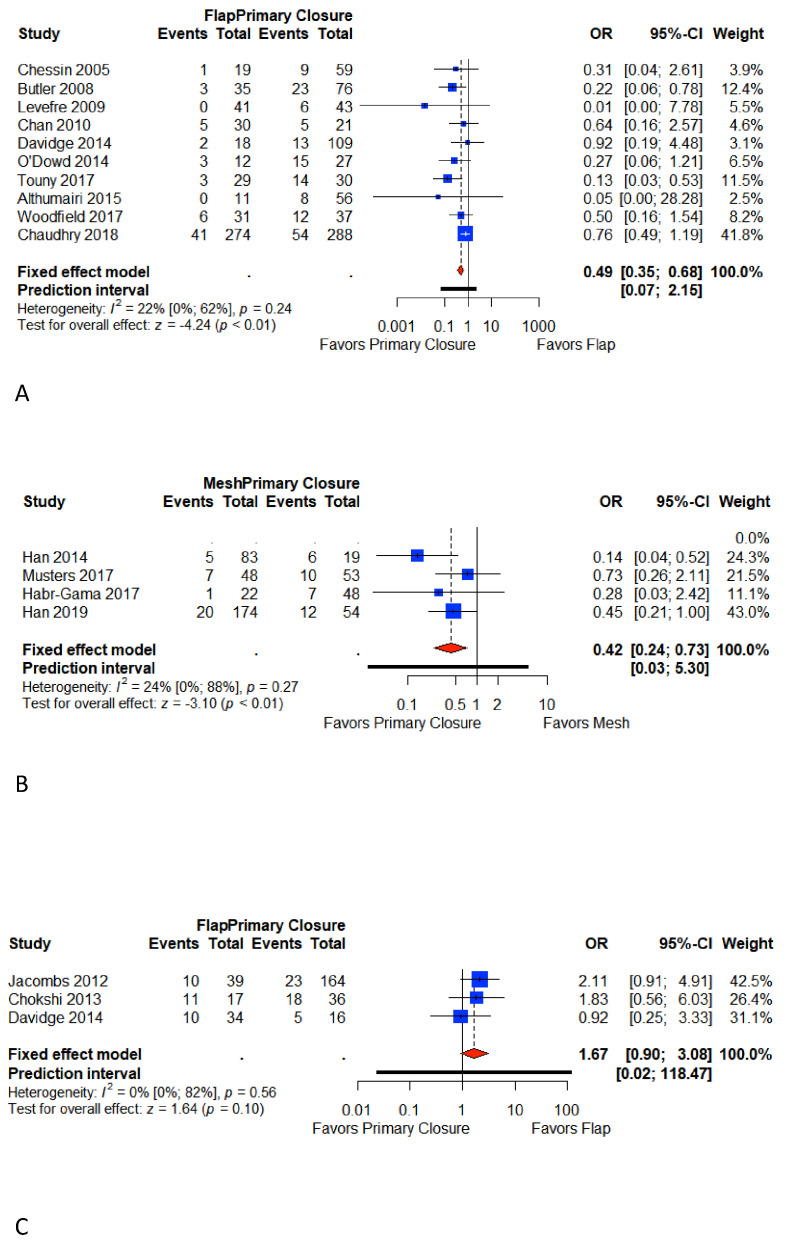
Forest plot for major perineal wound complications between patients with primary closure and with flap closure (**A**); forest plot for major perineal wound complications between patients with primary closure and with mesh closure (**B**); forest plot for major perineal wound complications between patients with primary closure and with flap closure in pelvic exenteration (**C**). Odds ratio: overall ratio of major perineal wound complications when comparing primary closure with flap closure. All weights are from random effects analysis. M–H, Mantel–Haenszel.

**Figure 4 cancers-13-00721-f004:**
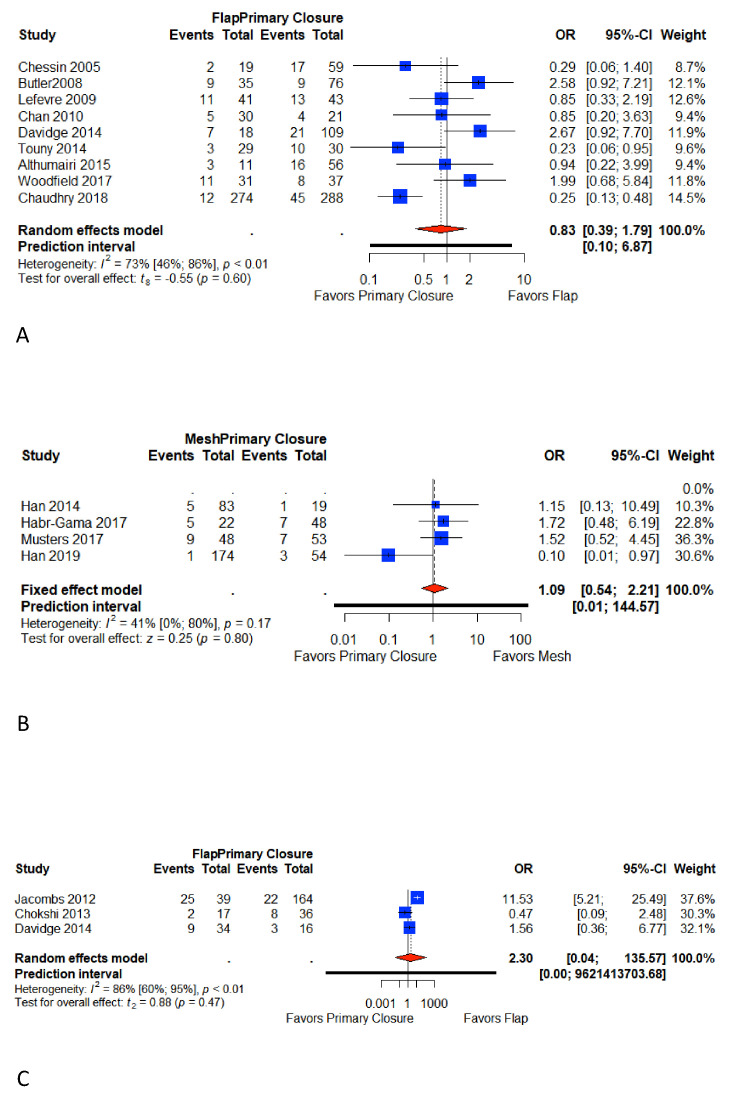
Forest plot for minor perineal wound complications between patients with primary closure and with flap closure (**A**); forest plot for minor perineal wound complications between patients with primary closure and with mesh closure (**B**); forest plot for minor perineal wound complications between patients with primary closure and with flap closure in pelvic exenteration (**C**). Odds ratio: overall ratio of minor perineal wound complications when comparing primary closure with flap closure. All weights are from random effects analysis. M–H, Mantel–Haenszel.

**Figure 5 cancers-13-00721-f005:**
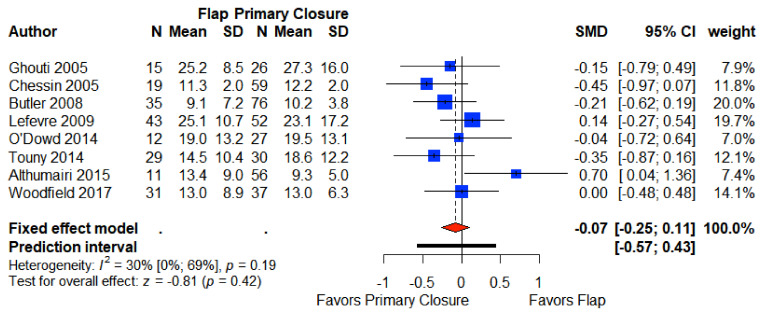
Forest plot for length of stay between patients with primary closure and with flap closure. Odds ratio: overall ratio of length of stay when comparing primary closure with flap closure. All weights are from random effects analysis. M–H, Mantel–Haenszel.

**Table 1 cancers-13-00721-t001:** Study description of the included studies.

Study (Author)	Year	Country	Design	Quality *	Disease	Patient (*n* = 2180)	Primary Closure ** (*n* = 1206)	Flap closure ** (*n* = 647)	Mesh Closure ** (*n* = 327)
Ghouti et al. [21]	2005	France	Retrospective cohort study	5	Anal cancer	41	15	26	-
Chessin et al. [6]	2005	USA	Retrospective cohort study	5	Rectal cancer and anal cancer	78	59	19	-
Butler et al. [7]	2008	USA	Retrospective cohort study	5	Rectal and anal cancer	111	76	35	-
Lefevre et al. [22]	2009	France	Retrospective cohort study	7	Anal cancer	95	52	43	-
Chan et al. [23]	2010	UK	Retrospective cohort study	4	Rectal cancer	51	21	30	-
Stelzner et al. [24]	2011	Germany	Retrospective cohort study	3	Rectal cancer	74	46	28	-
Jacombs et al. [33]	2012	Australia	Retrospective cohort study	5	Rectal cancer	203	164	39	-
Chokshi et al. [32]	2013	USA	Retrospective cohort study	5	Rectal cancer, anal cancer, and other cancers	53	36	17	-
Davidge et al. [25]	2014	Canada	Retrospective cohort study	4	Rectal cancer	177	125	52	-
O’Dowd et al. [27]	2014	Ireland	Retrospective cohort study	8	Rectal cancer	39	27	12	-
Touny et al. [28]	2014	Egypt	RCT	9	Rectal cancer	60	30	30	-
Althumairi et al. [26]	2015	USA	Retrospective cohort study	4	Rectal cancer	67	56	11	-
Woodfield et al. [29]	2017	New Zeland	Retrospective cohort study	3	Rectal cancer, anal cancer, and other cancers	68	37	31	-
Chaudhry et al. [30]	2018	USA	Retrospective cohort study	5	Rectal cancer, anal cancer, and other cancers	562	288	274	-
Han et al. # [31]	2014	China	RCT	9	Rectal cancer	102	19	-	83
Musters et al. # [10]	2017	Netherlands	RCT	9	Rectal cancer	101	53	-	48
Habr-Gama et al. # [18]	2017	Brazil	Retrospective cohort study	6	Rectal cancer	72 ***	48	-	22
Han et al. # [9]	2019	China	Retrospective cohort study	6	Rectal cancer	228	54	-	174

* Newcastle–Ottawa quality assessment scale. ** Patient included in analysis. *** 2 patients had a flap closure. # Mesh type: Han et al. [9,31], Human acellular dermal matrix (Ruinuo, Qingyuanweiye Bio-Tissue Engineer- ing Ltd., Beijing, China); Musters et al. [10], Strattice ®® LifeCell non-crosslinked dermis porcine; Habr-Gama et al. [18], Bio-A®® WL Gore synthetic bioabsorbable.

**Table 2 cancers-13-00721-t002:** Pooled baseline characteristics of study population with flap closure and primary closure; mesh closure and primary closure.

	Flap and Primary Closure (*n* = 1679)			Mesh and Primary Closure (*n* = 501)		
	Flap (*n* = 647) *	Primary Closure (*n* = 1032)	Total (*n* = 1679)	Mesh (*n* = 327)	Primary Closure (*n* = 174)	Total (*n* = 501)
Sex						
Male	463 (77%)	716 (69%)	1179 (70%)	164 (37%)	67 (38%)	319 (63%) **
NR	35 (5%)	76 (7%)		312 (32%)	67 (38%)	0
Disease ***						
Rectal cancer	256 (40%)	623 (60%)	879 (53%)	327 (100%)	174 (100%)	501 (100%)
Anal Cancer	88 (13%)	72 (7%)	160 (9%)	0	0	0
Other cancers	11 (2%)	0	11 (0.5%)	0	0	0
NR	292 (45%)	337 (33%)	629 (37.5%)	0	0	0
Neoadjuvant therapy						
None	141 (22%)	220 (21%)	361 (22%)	85 (26%)	35 (20%)	120 (23%)
Short course (25 Gy)	0	0	0	182 (56%)	49 (28%)	231 (47%)
Long course (40–60 Gy)	381 (59%)	541 (52%)	922 (55%)	60 (18%)	90 (52%)	150 (30%)
Chemotherapy	305 (46%)	455 (43%)	760 (45%)	232 (70%)	129 (75%)	361 (72%)
NR	125 (19%)	283 (27%)	398 (23%)	0	0	0
Type of resection						
APR	460 (71%)	749 (73%)	1209 (72%)	3 (1%)	47 (27%)	50 (9%)
ELAPE	97 (15%)	67 (6%)	164 (10%)	324 (99%)	127 (73%)	451 (91%)
Pelvic exenteration	90 (14%)	216 (21%)	306 (18%)	0	0	0
NR	0	0	0	0	0	0

* All patients underwent VRAM reconstruction except 35 gracilis flaps and 3 gluteus flaps. ** 2 out of 4 studies did not provide sex details between the two groups (Han 2014 [31]; Habr-Gama 2017 [18]). *** Choksi et al. [32] did not provide details on disease between squamous cell carcinoma and rectal cancer.

## Data Availability

The data presented in this study are all available in the present article and Appendix A included in the present manuscript.

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
