# Peer review of "Perineal Wound Closure Following Abdominoperineal Resection and Pelvic Exenteration for Cancer: A Systematic Review and Meta-Analysis"

_cancers, 2021, doi:10.3390/cancers13040721_

Round 1

Reviewer 1 Report

This is an interesting article regarding perineal wound closure after Miles procedure and pelvic exenteration.

The topic is interesting and the authors included all the updated studies.

The methodology of the study is correctly developed as well as the statistical analysis and description of the results.

A minimal editing of the English-language is useful to further improve the manuscript.

Author Response

Reviewer 1

This is an interesting article regarding perineal wound closure after Miles procedure and pelvic exenteration.

The topic is interesting and the authors included all the updated studies.

The methodology of the study is correctly developed as well as the statistical analysis and description of the results.

A minimal editing of the English-language is useful to further improve the manuscript.

RESPONSE: We thank the reviewer for their positive evaluation of our work. The whole manuscript has been edited and corrected again by a native English speaker (including correction of  numerous misspelling and language mistakes).

Reviewer 2 Report

This is a very well written and conducted Meta-analysis of closure methods for perineal wounds after abdominoperineal resection and pelvic exenteration. The topic is relevant and the methodology used conforms to modern standards. There are enough publications in the subgroups of methods analyzed to provide a relevant guidance. I have only two suggestions that could improve the manuscript:

  1. The proportion of ELAPE-cases for the treatment entities in groups A and B is a confounder. Is it possible to determine those proportions from the studies included?

  2. Different types of mesh used for closure of perineal wounds differ considerably in biomechanical properties. There is for example important differences between crosslinked and non-crosslinked biological meshes. It would be valuable if the type of mesh used in the included studies were listed.

Author Response

Reviewer 2

This is a very well written and conducted Meta-analysis of closure methods for perineal wounds after abdominoperineal resection and pelvic exenteration. The topic is relevant and the methodology used conforms to modern standards. There are enough publications in the subgroups of methods analyzed to provide a relevant guidance. I have only two suggestions that could improve the manuscript:

1-The proportion of ELAPE-cases for the treatment entities in groups A and B is a confounder. Is it possible to determine those proportions from the studies included?

RESPONSE: We agree and provide the detailed number of ELAPEs for each study in the supplemental data (Table 8,9,10 and 11) and summarized in Table 2. However, we were unable to obtain details of complications between APR and ELAPE.

2-Different types of mesh used for closure of perineal wounds differ considerably in biomechanical properties. There is for example important differences between crosslinked and non-crosslinked biological meshes. It would be valuable if the type of mesh used in the included studies were listed.

RESPONSE: We agree and have reported the different types of mesh used in each study in the table1.  Han et al Human acellular dermal matrix (Ruinuo, Qingyuanweiye Bio-Tissue Engineer- ing Ltd, Beijing, China); Musters et al Strattice® LifeCell non-crosslinked dermis porcine; Habr-Gama et al Bio-A® WL Gore synthetic bioabsorbable.

Reviewer 3 Report

Buscail and colleagues present a systematic review and meta-analysis regarding perineal wound complications following APR and PE. Perineal wound complication are an important source of morbidity (and rarely mortality) following rectal and anal cancer surgery and recommendations for wound closure are seldom based on scientific evidence. Therefore I think the manuscript can provide important insight on the topic and should be published.

Nevertheless, some minor issues should be corrected prior to publication:

I had to read several times to understand the three groups you chose for comparison (PC vs. flap/PC vs. mesh and PC vs. flap in PE). You should explain your choice more plain, especially group C.

p3 second paragraph: It is not clear how you build a Southhampton score, if it was not mentionend in the primary study. Consider another supplementary table in which you display your choice of classification of Southhampton score for each study.

p6 You state that "the homogeneity of the studies... strengthens these findings". If I look at table 2 I do not find both groups distributed homogeneously. This is, of course, due to the different disease, stage and pretreatment, which imply a certain therapy in mostly retrospective cohort stiudies. You should at least discuss this in more detail. On p6l280 you mean only including VRAM as "homogeneity of therapy"?

Author Response

Reviewer 3

Buscail and colleagues present a systematic review and meta-analysis regarding perineal wound complications following APR and PE. Perineal wound complication are an important source of morbidity (and rarely mortality) following rectal and anal cancer surgery and recommendations for wound closure are seldom based on scientific evidence. Therefore I think the manuscript can provide important insight on the topic and should be published.

Nevertheless, some minor issues should be corrected prior to publication:

I had to read several times to understand the three groups you chose for comparison (PC vs. flap/PC vs. mesh and PC vs. flap in PE). You should explain your choice more plain, especially group C.

RESPONSE: we agree with the reviewer and have modified the text as follows:

To reach optimal homogeneity and comparability, we divided the studies into three groups as follows: Group A studies comparing primary and flap closure in APR and ELAPE, Group B studies comparing primary and mesh closure in in APR and ELAPE, and Group C studies comparing primary and flap closure in pelvic exenterations only (page 3, second paragraph). 

p3 second paragraph: It is not clear how you build a Southhampton score, if it was not mentionend in the primary study. Consider another supplementary table in which you display your choice of classification of Southhampton score for each study.

RESPONSE: We agree and when the Southampton score was not available we used the descriptive items of the score to determine major and minor complications. We have therefore implemented the supplemental table 2 as follows:

Grade

Definition

Appearance

0

Normal healing

I

Normal healing with mild bruising or haematoma

A – some bruising
B – considerable bruising
C – mild erythema

II

Erythema plus other signs of inflammation

A– at one point
B – around sutures
C – along wound
D – around wound

III

Clear or haemoserous discharge

A – at one point only (<2cm)

B – along wound (> 2 cm)
C – large volume
D – prolonged (> 3 days)

IV

Pus

A – at one point only (<2cm)

B – along wound (> 2 cm)

V

Deep or severe wound infection with or without tissue breakdown; haematoma requiring aspiration

p6 You state that "the homogeneity of the studies... strengthens these findings". If I look at table 2 I do not find both groups distributed homogeneously. This is, of course, due to the different disease, stage and pretreatment, which imply a certain therapy in mostly retrospective cohort stiudies. You should at least discuss this in more detail. On p6l280 you mean only including VRAM as "homogeneity of therapy"?

RESPONSE: We agree and modified the discussion as follow :

The homogeneity of surgical strategies (e.g VRAM flap and exclusion of pelvic exenteration) of the studies included here strengthens these findings. The flap donor site complications are one important aspect but are not discussed here, as we have focused our analyses on perineal wound healing (page 6)

 […] Indications for perineal resection in groups A and C included patients with rarer cancers of the anus, rectum and other neoplasias with less homogeneous neo-adjuvant treatments (page 7).